# Does the Reflection of Foci of Commitment in Job Performance Weaken as Generations Get Younger? A Comparison between Gen X and Gen Y Employees

Ahmet Alkan Çelik [1], Mert Kılıç [2,*], Erkut Altındağ [3], Volkan Öngel [1] and Ayşe Günsel [2]

[1] Economics Department, Faculty of Economics and Business Sciences, Beykent University, İstanbul 34000, Turkey; alkanc@beykent.edu.tr (A.A.Ç.); volkanongel@beykent.edu.tr (V.Ö.)
[2] Management and Organization Department, Faculty of Economics and Business Sciences, Kocaeli University, Kocaeli 41380, Turkey; agnsel@gmail.com
[3] Management Department, Faculty of Economics and Business Sciences, Beykent University, İstanbul 34000, Turkey; erkutaltindag@beykent.edu.tr
* Correspondence: mertkillic@gmail.com

**Abstract:** Today's organizations increasingly recognize the fact that employees and employee performance are essential intangible assets that should be effectively managed. Affective commitment (AC) is a widely recognized antecedent of sustainable job performance. However, achieving AC has become a great challenge in general and has been especially difficult since the beginning of the pandemic because almost all companies asked their employees to stay at home and work remotely in an isolated manner. Today, many different generations work side by side, contrary to the past, when generational mixing was very rare. Many differences exist among these employee generations, which determine their feelings towards authority and organization. Accordingly, this paper aims to clarify generational differences in the interrelationships among AC and sustainable job performance between Gen X and Gen Y employees. As remote working structure limits the interactions that employees have with their supervisor, fellow employees, and the organization, we decided to use the foci of commitment: affective commitment to the organization, affective commitment to the supervisor, and affective commitment to coworkers. Based on data from 416 post-graduates of Beykent University and using the PLS-SEM technique, we find that commitment to the supervisor and commitment to the organization are positively associated with job performance. Moreover, the findings reveal that the impact of the relationship between commitment to the supervisor and job performance is weaker for Gen Y than for Gen X.

**Keywords:** job performance; foci of commitment; generations





## 1. Introduction

In today's global and hyperdynamic environment, organizations face various challenges in achieving sustainability and success [1]. In particular, due to the unexpected worldwide spread of COVID-19 at the beginning of 2020, almost all companies asked their employees to stay at home and work remotely in an isolated manner, disrupting the structure of the workplace and weakening the connection between organizations and employees [2]. Moreover, reduced business led to furloughs and closures. For this reason, employees of these companies have had to cope with an additional source of stress or pressure while staying at home for a longer period of time [3]. In order to effectively alleviate these pressures and achieve organizational survival and success, organizations increasingly recognize the fact that employees and their performance are essential intangible assets that should be effectively managed [4]. Individual job performance is generally described as the total contribution of the employee's implemented tasks and actions to the firm's aim and future success. There are no elements of sustainable team performance, firm

financial performance, industry performance, or gross domestic product (GDP), except the sustainability of business performance [5].

An abundance of studies on job performance have addressed the facilitator or antecedent role of organizational commitment (OC) in general and AC in particular (e.g., [6–12]). Affective commitment refers to the emotional attachment that employees have to an organization [13,14]. As emotionally and psychologically based AC is essential for achieving organizational tasks, AC strengthens the relationship between the organization and employees [15]. We have recently witnessed more attention on AC from organizational researchers due to the changes occurring in today's modern business environments and business practices all around the globe [16]. For instance, in the workplace, there are many different generations, including Millennials, Gen Xers, and Gen Yers, working closely side by side, with both people who are as young as their children and as old as their parents, contrary to the past, when generational mixing was very rare [17]. There are fundamental differences in perceptions, stereotypes, and personalities between employee generations. These differences, real or imagined, determine employees' feelings towards authority and organization, and they involve potential areas of conflict across generations [18]. It is of utmost importance to understand and meet the expectations of today's workforce, which is characterized by diversity in general and generational diversity in particular, to obtain commitment and high performance from employees, which can lead to the long-term survival and success of companies [19]. Hence, Nelson [20] underlines the importance of generational differences and suggests that their impact on levels of affective commitment should be examined.

Accordingly, this paper aims to reveal the effects of AC on sustainable job performance. Moreover, to contribute to filling the existing research gap in the literature on OC, behavior, and generational differences, we also attempt to clarify the role of generational differences in the interrelationships between AC and sustainable job performance [16]. For this purpose, we decided to use the foci of commitment: affective commitment to the organization, affective commitment to the supervisor, and affective commitment to coworkers. Indeed, the traditional understanding of commitment is that employee attachment involves "the relative strength of an individual's identification with and involvement in a particular organization," not a person [17]. However, several scholars have begun to distinguish between foci and bases of commitment [21,22]. Foci of commitment represent the individuals and groups in which an employee is emotionally involved, such as commitment to professions, top management, supervisors, coworkers, and customers, based on different motivational processes [23–26]. Based on the fact that a remote working structure limits the interactions that employees have with their supervisor, fellow employees, and the organization [27], we inferred that deeper insight could be gained by examining the reflection of AC in sustainable job performance, particularly in remote working conditions during the pandemic. For this purpose, we consider job performance as a two-dimensional construct—task performance and contextual performance—following Sackett's [28] study. We attempt to advance the OB literature by exploring the antecedent role of foci of affective commitment in sustainable job performance.

Furthermore, we also aim to enrich the OB literature by exploring the role of generational differences—between Gen X and Gen Y—on the relationships between foci of commitment and individual job performance, specifically in pandemic conditions. Gen Yers are believed to be the first of their kind, as this generation was born into a wired world that allows them to be connected 24/7. In this sense, they are called the internet generation [29].

The fact that a remote working structure limits the interactions that an employee has with a supervisor, fellow employees, and the organization as a whole raises the question of how this working structure influences affective commitment to the organization.

The expectations or reactions of Gen Yers regarding online/remote working are expected to differ from those of Gen Xers. Hence, this study is substantially guided by the following two important research questions: (i) How do effective foci of commitment

affect commitment to sustainable job performance? (ii) Do the effects of foci of affective commitment on sustainable job performance differ across generations? Interestingly, even though AC is a well-studied concept in OB and there are many studies showing its antecedent role in job performance, the interrelationships among foci of commitment and job performance from a generational perspective are relatively unexplored in the literature in general and in developing countries, such as Turkey, in particular. To the best of our knowledge, no methodological framework for such a holistic approach has been developed yet. From a theoretical perspective, this study contributes to OB theory by revealing how the effects of foci of affective commitment on job performance vary due to generational differences. From a practical standpoint, the proposed model (Figure 1) can help managers, particularly HRM managers, to enhance their understanding of how to increase the AC and the long-term performance of employees, particularly employees of Gen Y, for organizational sustainability.

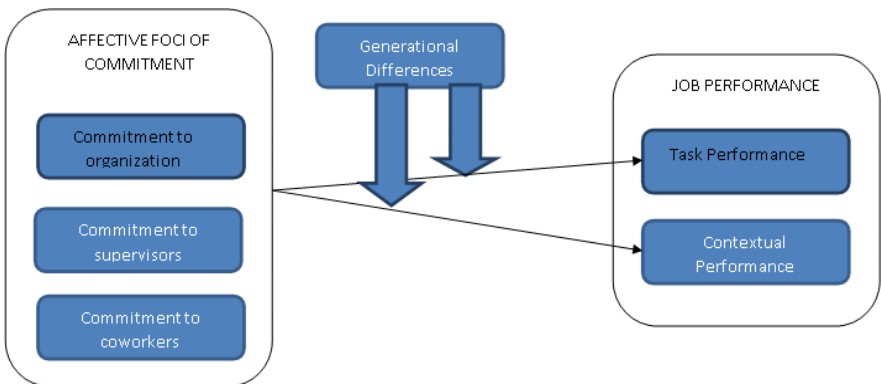

**Figure 1.** Research model.

## 2. Literature Review

### 2.1. Job Performance

Job performance is of interest and value to any organization since its ultimate success or failure mainly depends on its employees' performance [30]. When employees have superior performance, it advances the organization and makes it successful and effective [31]. Performance refers to an intentional act of individuals, resulting in action and guided by outputs, and is the result of an initial motivation or a conscious purpose. Job performance is influenced by economic, social, cultural, and demographic conditions, in addition to work conditions. Moreover, several factors (e.g., employee features, organizational characteristics, and business environment) and different personalities can affect job performance [32–34]. If job performance is expected to last for a long time, that is, if job performance is expected to be sustainable, then the coordination of internal and external motivations for performing core work activities is required. Sustainable job performance refers to continuously meeting or exceeding performance objectives. To realize this continuity, all possible determinants of job performance should be considered together in a holistic manner [35].

Job performance is also considered to be an element that can be used for all strategies and interventions regarding firm performance. Most of these strategies emphasize performance-increasing issues, such as training and development, motivation, recruitment, and selection. However, from another perspective, fellow workers focus on removing constraints that prevent individuals from contributing to predetermined corporate goals and providing individuals with advanced opportunities for corporate contributions.

Many studies in the extant literature show a strong positive relationship between job performance and organizational aspects (e.g., [30,36,37]). Job performance is defined as the total expected value to the organization of individual functional segments in which an employee has been operating for a predetermined period of time. An essential concept

in this definition is that job performance is a characteristic of employee behavior. In particular, it is a cumulative feature of multiple, distinct behaviors that occur over a given time. The second important concept is that behavior related to performance refers to its expected value to the organization. This is a variable that distinguishes between sets of behaviors exhibited by different employees and between groups of behaviors exhibited by the same person at different times. The distinction is based on how much (in total) behaviors contribute to or reduce organizational effectiveness. In a word, the variance in performance is the variance in the expected organizational value of the behavior. There are cases where employees with higher performance, especially in for-profit companies, increase the company's value [38].

Task performance refers to the quality and quantity recognized by a firm's official human resource management system [39,40]. The most basic requirements for categorizing performance are divided into three groups. The first is to join an organization and continue being employed there. The second is to reliably meet or exceed the performance standards set by corporate roles. Finally, the third is to be innovative and spontaneous. Collaborating with other members is to transcend predetermined roles to carry out actions such as protecting the organization from crises, offering improvement and renewal suggestions, devoting himself/herself to development, and making the organization a learning organization [41,42]. The theory focuses on the question of why some people perform better than others in their job duties. If employees are equal in skill and knowledge, then the reasons that some perform better than others are examined. Task performance considers motivation from a first-order perspective, with an emphasis on an instantaneous level of explanation of individual differences in task performance [43,44]. In terms of content, the most important way to increase employees' job performance is to eliminate stress. Work is an important aspect of life for most individuals and has serious spillover effects on personal outcomes, such as workplace successes or failures, good or bad performance, psychological health, and family life [45,46].

Borman and Motowidlo call these task-specific roles "task performance". In addition to task performance, roles that exceed the expected task requirements refer to "contextual performance" [47]. The concept of task performance is directly related to the core capabilities of the organization resulting from successfully executing specific and technical processes, maintaining technical requirements, and providing services. In addition, contextual performance includes employee behaviors that do not support core competencies themselves. It supports the broader organizational, social, and psychological environment in which the technical core task must be implemented. Moreover, contextual performance does not specifically apply to the assigned tasks of employees. However, it is also an essential part of job performance, whereas task performance behaviors are defined according to role requirements and should be defined by a comprehensive analysis of the job. The primary purpose of both mentioned performance types is to maintain the highest level of employee motivation and provide employees with ways to conduct business that will enable them to continue their activities. Especially in learning organizations, the reference point that an institution is based on is the past average performance. From this point of view, task performance and contextual performance are important output sources that can be used by companies to make objective measurements [48,49]. Indeed, objectivity and reliability for performance measurement are increasingly debated. Although self-reported performance measures are subjective and create some limitations due to their subjective nature, many studies have reported a strong correlation between objective and self-reported performance measures [50–57]. According to Anjum et al. [58], the deployment of self-reported measures may only become a disadvantage if real conditions are misjudged. Moreover, in this paper, we aim to examine job performance from a long-term perspective, that is, "sustainable" performance. Since this long-term performance or continuity is based on internal motivations and isolated from the other evaluation biases, subjective performance appraisals—self-reports—may provide broader input on employee performance than objective tools [59]. Hence, we decided to use self-reported measures of

job performance, which enables us to evaluate job performance in two dimensions: task performance and contextual performance.

### 2.2. Foci of Commitment

Organizational commitment represents employees' attachment to the organization in terms of their emotions, attitudes, and behaviors [60]. According to Allen and Meyer [61], organizational commitment is a psychological link between employees and the organization, minimizing the employee's probability of quitting. Porter et al. [62] conceptualize organizational commitment as involving three main components:

- A powerful belief in and internalization of the organization's goals;
- A willingness to apply significant effort for the organization;
- A clear desire to continue being a member of the organization.

Organizational commitment is an attitude expressed through the loyalty of employees to the organization. It is a determinant of employee effort and performance [63].

In OB literature, there are two mainstream classes of organizational commitment theories: behavioral commitment theories, originally developed by Becker [64] and Salancik and Pfeffer [65], and attitudinal commitment theories, developed by Kanter [66], Etzioni [67], Mowday, Steers, and Porter [68], O'Reilly and Chatman [69], Penley and Gould [70], Wiener [71], and Allen and Meyer [72]. Behavioral commitment highlights employee behaviors that benefit the organization [73]. As suggested in Salancik and Pfeffer's [65] social information processing model, these behaviors are the indicators of employees' attitudes. Attitude provides a subjective framework for individuals about a particular situation or object. Attitude is perhaps the most frequently used variable to identify what underlines individuals' judgments and behaviors. On the other hand, organizational commitment arises as the most common attitude used to characterize employees' relationships with the organization [74].

As a highly studied concept, organizational commitment has many classifications and categorizations. Among the many constructs, Allen and Mayer's [72] three-dimensional model, involving affective commitment, normative commitment, and continuance commitment, seems to have wide acceptance [75]. First, affective commitment refers to the employee's emotional attachment to, identification with, and involvement in the organization. Second, normative commitment emphasizes the sense of moral obligation of employees toward the organization. Finally, continuance commitment refers to the employee's desire to maintain organizational membership because of the costs of quitting [76]. Many authors consider affective commitment to be the real commitment among the three categories since it represents a genuine and willing attachment.

Although organizational commitment is a well known and thoroughly studied concept in OB studies, foci of commitment have emerged as a relatively new facet of commitment that underlines the focus points toward which commitment is directed. The central focus point, of course, is the organization; however, the extant literature also suggests other foci, such as commitment to the profession or career, commitment to top management, commitment to supervisors, commitment to coworkers, commitment to the union, commitment to the workgroup, or commitment to customers [77]. Among the many foci, these three are considered to be especially important for sustainable competitive advantage, as they are in the organization's inner circle.

Commitment to the supervisor is defined as the psychological attachment of employees to their supervisor, and it is directly related to the behaviors and attitudes of the supervisor or leader [78]. The first studies on commitment to supervisors were conducted based on the research of Reichers [24]. Reichers [24] argued that employees who are highly devoted to their colleagues and supervisors pay great attention to sincere relationships and friendship bonds. They prefer to work in an environment surrounded by people with whom they can share mutual experiences and cooperate. Loyalty between friends is the ultimate goal for individuals with this type of personality. It is particularly difficult for them to quit their job since it also means abandoning friends [24].

Commitment to coworkers refers to identifying with other colleagues and having a sense of loyalty to them. Relationships with colleagues (i.e., one's supervisor and coworkers) play an essential role in determining emotional attachments to the organization [79]. Commitment to coworkers can sometimes emerge as a means to gain certain benefits, and other times, it may be the benefit.

In summary, commitment to coworkers is an individual's identification with and loyalty to other employees [80]. Affective commitment is positively associated with high-quality leader–member exchanges [81] and interpersonal relationships among coworkers [82]. There is also evidence that workers who tend to form strong interpersonal relationships tend to have increased commitment to coworkers, ultimately resulting in an affective commitment to the organization (e.g., [82]).

*2.3. The Reflection of Foci of Commitment on Job Performance*

A review of the literature published in the last three decades shows that numerous studies provide empirical evidence regarding the effects of organizational commitment on job performance. Chronologically, Meyer et al. [6] first found that job performance positively correlated with affective commitment and negatively correlated with continuance commitment. The findings of Meyer et al.'s [6] research revealed the importance of distinguishing between need-based commitment and desire-based commitment. This result was interpreted as supportive of the organizational efforts of employees to strengthen emotional commitment. Riketta's [83] meta-analysis yielded some striking results as well. A meta-analysis was conducted to estimate the relationship between attitudinal organizational commitment and job performance and identify moderators of this correlation. Ninety-three published scientific studies were examined, and one hundred eleven samples suitable for the research were included in the content. The most important of the results was that the additional task commitment was particularly crucial for white-collar workers. In 2007, Hunter and Thatcher looked at commitment, stress, and performance in their study. They found that employees with higher levels of affective commitment had higher job performance.

By the 2010s, the scope of studies on commitment foci had broadened. Notably, researchers began to analyze the relationship between the elements of commitment and more popular concepts, such as organizational climate, culture, work stress, and organizational justice. The concept of commitment was evaluated at different levels of relationships and with varying analysis techniques, such as moderator variables. Tolentino's study on academic staff working at universities in 2013 revealed that attendance, commitment, and attendance dimensions directly affected job performance. Sharma and Dhar [84] found that affective commitment had a substantial impact on the job performance of health workers in terms of both task performance and contextual performance. The common element of the studies mentioned above is that the foci of commitment in general and foci of affective commitment in particular are at the center of the research in various sectors and are measured with different variables. Thus, we expect AC to increase employees' job performance, both task performance and contextual performance. Accordingly:

**Hypothesis 1 (H1).** *AC towards the supervisor, coworkers, and the organization has positive and significant effects on task performance.*

**Hypothesis 2 (H2).** *AC towards the supervisor, coworkers, and the organization has positive and significant effects on contextual performance.*

*2.4. Generational Differences as a Moderator of the Relationship between Foci of Commitment and Job Performance*

Today, generational differences are an essential form of diversity in business life that must be recognized as having impacts on organizations in the long term [19]. Karl Mannheim [85] first proposed the concept of "generations" more than half a century

ago. Generations denote groups of individuals or cohorts derived from their common experiences at similar or close ages. Mannheim argues that membership in a generation can dictate "certain patterns of behavior, emotion, and thought" in his *Theory of Generations* [85]. The basic principle behind the concept of generations is that individuals are affected or even shaped by common experiences, such as historical events, socioeconomic conditions, and cultural phenomena, throughout adolescence and early adulthood [86–88]. Common experiences and activities shared by individuals of a certain age at a given time create various similarities (e.g., political orientations, reactions, attitudes, general tendencies) among group members [89]. Although Lyons et al. [90] argued that any scholar studying generational cohorts will face an age–period–cohort confounder, all members of the same age group are, of course, not identical. These differences may be attributed to many causes other than the age–period–cohort factor [91]. For instance, Derecskei et al. [92] argued that the differences in personality and other psychological characteristics and the environment in which the person is socialized make each group diverse. Moreover, age differences (developmental psychology) should also be considered, since members of Generation X may have become apathetic, whereas Generation Y individuals are ambitious and active. Derecskei et al. [92] strongly emphasized the possibility that members of Generation X were also full of ambition at the same age, when they started to explore the job market. Despite these other determinants of employee similarities or differences, many OB studies have observed a number of significant stereotypes with respect to generational cohorts (e.g., [19,92–94]). Thus, when employed as a lens to examine differences in employee attitudes, behaviors, and expectations, generational differences emerge as a useful tool [19].

The generational theory considers each generation to span around 20–25 years. Different generations are identified and classified via their year of birth. Each generation has its own experiences, resulting in the emergence of similarities among its members, such as mental models, values, beliefs, attitudes, or traits [95]. The extant literature provides many classifications for differentiating one generation from another. The classification of Lyons and Kuron [96], which has gained wide acceptance, assumes four generations to be actively participating in professional life:

1. Baby Boomers (BB): those born between 1946 and 1961;
2. Gen X: those born between 1962 and 1979;
3. Gen Y: those born between 1980 and 2000;
4. Gen Z: those born between 2000 and 2020 [95].

In this paper, we also apply Lyons and Kuron's [96] generational classification, as it is considered to be appropriate for Turkish society (e.g., [97,98]). Turkey witnessed an important coup in 1980 and the introduction of the internet in 2000. Since Kupperschmidt [99] defines a generation as "an identifiable group (cohort) that shares birth years, age location, and significant life events at critical developmental stages (times) . . . " (p. 66), Lyons and Kuron's [96] classification seems to meet the criteria of (1) significant life events and (2) critical development stages. Of course, based on that classification, there are now four generations actively participating in professional life. However, as previously mentioned, Gen X (born 1962–1979) and Gen Y (born 1980–2000) are currently predominant in the workforce [100]. Gen Xers are defined as skeptical and individualistic. They and their personal lives are more important than work, so they are less likely to sacrifice for a career. They are less hierarchical and more sophisticated than BBs [99,101]. Gen Yers are not as good, or are probably even worse, with hierarchy in the workplace. They are considered more self-confident and even more narcissistic than Gen Xers; they have difficulty with authority. On the other hand, they are more career- and success-oriented [95,102–104].

There are unavoidable differences in the perceptions, stereotypes, and personalities between these employee generations. Those differences, real or imagined, determine employees' feelings towards authority and the organization, and they involve potential areas of conflict among generations [18]. The OB literature includes many studies regarding the work-related consequences of generational differences in various aspects, such as commitment, satisfaction, performance, risk-taking, and leadership style [89]. Some studies

examining generational differences have revealed significant differences in commitment. For instance, Nelson [20] underlines the impact of generational differences on affective commitment. According to Lub et al. [105], Gen Y employees present lower commitment and higher turnover intentions than Gen Xers. Gen Yers are more easily seduced by better opportunities and have no problems quitting, which leads to a turnover problem. Twenge [106] associates Gen Y behavior with their selfish and career-oriented nature. Thus, the business world is experiencing a decrease in psychological attachment to their organization and thus a decline in affective commitment. Affective commitment has been proved to provide many benefits, such as lower absenteeism, lower turnover, and even higher job performance [107]. Moreover, Gen Y is commonly considered the "Net" generation. They grew up with greater access to technology and are more prone to taking "digital" breaks from work. Furthermore, autonomy is another important determinant of commitment and performance for Gen Y [108,109]. Hence, we expect the nature of the relationship between affective commitment and job performance to be different between Gen X and Gen Y employees, particularly in pandemic conditions characterized by remote working based on online technologies. Assuming that Gen Yers are more narcissistic and technology- and self-oriented with a lack of attachment to their organizations, we expect that the impact of the relationship between organizational commitment and job performance—task performance and contextual performance—will be lower for Gen Y than for Gen X employees.

Accordingly:

**Hypothesis 3 (H3).** *Relationships between affective commitment towards the supervisor, coworkers, and the organization and task performance are moderated by generational differences such that commitment–performance relationships are weaker for Gen Y than for Gen X.*

**Hypothesis 4 (H4).** *Relationships between affective commitment towards the supervisor, coworkers, and the organization and contextual performance are moderated by generational differences such that commitment–performance relationships are weaker for Gen Y than for Gen X.*

## 3. Research Design

### 3.1. Measures

In order to test the hypotheses mentioned above, multi-item scales obtained from previous empirical studies were used for the measurement of constructs. Each construct used to assess generational differences was measured using a 5-point Likert scale, ranging from "strongly disagree" (1) to "strongly agree" (5). Generation was determined by directly asking for the age of the respondent; thus, it was transformed into a categorical scale, in which 1 represents Generation Y and 2 represents Generation X.

To measure organizational commitment, the foci of commitment scale developed by Wasti and Can [110] were used. These researchers distinguish both different forms (affective and normative) and different foci (organization, supervisor, coworkers) of commitment. Since many organizational behavior (OB) studies have considered affective commitment to be the actual form of commitment that reflects the willing commitment of employees [111], we used the affective form of the foci of commitment scale. Thus, we used the foci of commitment scale involving the dimensions of affective commitment to the supervisor, affective commitment to coworkers, and affective commitment to the organization, with five items for each dimension. "Working with my supervisor has a great deal of personal meaning for me" (commitment to supervisor), "I really feel as if my coworker's problems are my own" (commitment to coworkers), and "I would be very happy to spend the rest of my career with this organization" (commitment to organization) are some examples of the items in the scale.

A performance scale of 9 items was adapted from Goodman and Syvantek [112] to measure task performance, while another performance scale of 7 items was adapted from Jawahar and Carr [113] to measure contextual performance. "Achieves the objectives of

the job" (task performance), "Makes innovative suggestions to improve the overall quality of the department" (contextual performance), and "Helps other employees with their work when they have been absent" (contextual performance) are some examples of the performance items.

### 3.2. Sampling

This paper aims to describe and analyze the mutual relationships among foci of commitment and sustainable job performance from the perspective of generational differences.

To empirically investigate the hypotheses, 782 post-graduates, identified from the Alumni Association of Beykent University records, who are actively working and have been employed at the same organization for at least three years and are located in Istanbul, were chosen as the target sample based on their accessibility. In the first stage, a total of 782 participants were contacted directly by phone, and the purpose of the study was explained to them in detail with the expected outputs. Of the 782 personnel contacted, 512 agreed to participate in the study. However, only 429 of these participants were able to successfully complete the survey. After a detailed evaluation, all incomplete or incorrectly filled in questionnaires were excluded from the study, and 416 responses remained for analysis. The findings obtained as a result of the research are based on data obtained from a suitable sample of 416. There were 248 Gen Y (59.6%) and 168 (40.4%) Gen X participants. Male participants accounted for 72.5% of the sample ($n = 301$), 66.3% of the participants were single ($n = 276$), 26.9% had a bachelor's degree ($n = 112$), 68.9% had master's degree ($n = 287$), and 4.1% had doctorates ($n = 17$).

### 3.3. Analysis

The PLS-SEM technique was used to test the model based on critical statistical factors. First, according to Fornell and Larcker [114], PLS is a technique that avoids many of the restrictive assumptions underlying maximum likelihood techniques. It provides robustness against inappropriate solutions and factor uncertainty. PLS-SEM does not make any distribution assumptions regarding indicators or error terms [115]. In fact, PLS is a latent variable modeling technique that includes multiple dependent structures and explicitly recognizes measurement error. Second, the PLS technique is insensitive to sample size considerations and is suitable for any sample size above 30, unlike covariance-based structural equation modeling techniques [114,115]. Chin et al. [116] stated in their study that power analysis is based on the part of the model that has the most significant number of predictors. Finally, this technique deals with both formative and reflective structures [115].

### 3.4. Measurement Validation

In this study, following Kleijnen, Ruyter, and Wetzels [117], we used reflective indicators for all our constructs. The validity assessment of reflective measurement models focuses on convergent validity and discriminant validity. For convergent validity, researchers need to examine the average variance extracted (AVE). An AVE value of 0.50 or higher indicates a sufficient degree of convergent validity. Composite reliability and Cronbach alpha values from 0.60 to 0.70 in exploratory research and values from 0.70 to 0.90 in more advanced stages of research are regarded as satisfactory [118]. Moreover, each indicator's reliability needs to be taken into account, whereby each indicator's absolute standardized loading should be higher than 0.60 [119]. For all measurements, the PLS-based CR was determined to be well above the threshold of 0.70. Cronbach alpha exceeded the threshold of 0.70, and AVE exceeded the threshold of 0.50 for all first-order structures.

Table 1 shows the correlation between all seven variables. This table also provides further evidence of discriminant validity. Discriminant validity refers to how the constructs empirically differ from each other. This technique also measures the extent of differences between the overlapping structure [120]. The overarching criterion for assessing discriminant validity is the Fornell–Lacker criterion [121]. This method compares the square root of AVE with the correlation of hidden structures. A latent construct should better explain the variance of its indicator than the variance of other latent constructs. Therefore, the square root of the AVE of each structure must have a greater value than the correlations with other hidden structures [120]. In the model, none of the intercorrelations of the constructs exceeded the square root of the AVE of the constructs (see Table 1; the square root values of AVE are given diagonally in the table). Finally, we evaluated convergent validity by inspecting the standardized loadings of the measures on their respective constructs. We found that all measures exhibited standardized loadings that exceeded 0.60 (see Appendix A).

**Table 1.** Correlations and descriptive statistics.

| Variables | ccw | cp | co | cs | tp |
|---|---|---|---|---|---|
| ccw | **0.821** | | | | |
| cp | 0.221 | **0.755** | | | |
| co | 0.487 | 0.285 | **0.828** | | |
| cs | 0.474 | 0.248 | 0.479 | **0.787** | |
| tp | 0.135 | 0.814 | 0.267 | 0.148 | **0.790** |
| CR | 0.912 | 0.902 | 0.897 | 0.864 | 0.930 |
| AVE | 0.674 | 0.570 | 0.686 | 0.619 | 0.624 |
| $\alpha$ | 0.879 | 0.873 | 0.857 | 0.800 | 0.915 |

Note: ccw: commitment to coworkers; co: commitment to organization; cs: commitment to supervisor; tp: task performance; cp: contextual performance; CR: composite reliability; AVE: average variance extracted; $\alpha$: Cronbach alpha.

### 3.5. Hypothesis Testing

The PLS (partial least squares) approach [122] and the bootstrapping resampling method were performed by using the SmartPLS 3.0 software program to estimate the interaction and indirect effects in addition to the main effects and to test the hypotheses and predictive power of our proposed model (see Figure 1). T-statistics were estimated for all coefficients based on their stability across subsamples to define the statistically significant links. The path coefficients and their associated t-values show the direction and impact of each hypothesized relationship.

Table 2 shows the results of hypothesis tests, including paths, betas, and significance levels. Regarding the effects of foci of commitment on sustainable individual performance, the findings demonstrate that commitment to the organization was significantly and positively correlated with both task performance ($\beta = 0.23$; $p < 0.01$) and contextual performance ($\beta = 0.19$ $p < 0.01$), and commitment to the supervisor was found to have a significant effect on task performance ($\beta = 0.13$; $p < 0.05$) and contextual performance ($\beta = 0.11$; $p < 0.05$), supporting H1a, H1c, H2a, and H2c. Surprisingly, the results provide no empirical evidence in support of a direct relationship between commitment to the supervisor, task performance, and contextual performance, or a relationship between commitment to the supervisor and task performance.

A two-step construction procedure was used to address the hypotheses about the moderating effects of generations, i.e., H3 and H4 [122]. The PLS approach is well known to allow for an explicit estimation of the standardized latent variable scores after saving the obtained results [95]. To eliminate the collinearity problem, the interaction terms were established using the product indicator approach [122], which entails standardizing the items of constructs and computing the interaction term by multiplying each item of one construct with all the items of the moderator. Here, each item of commitment to the supervisor, commitment to coworkers, and commitment to the organization and generation differences were standardized. Following this procedure, the standardized question

items were multiplied. The results demonstrate negative interaction effects between commitment to the supervisor and task performance (β = −0.12; *p* < 0.05) and contextual performance (β = −0.18; *p* < 0.05), so H3a and H4a are supported. However, the results provide no empirical evidence in support of a statistically significant interaction effect between commitment to coworkers, commitment to the organization, and any individual performance dimensions.

**Table 2.** Results of hypothesis testing.

| Relationships | | | Path Coefficient (β) | Subhypotheses | Subresults | Hypotheses | Results |
|---|---|---|---|---|---|---|---|
| cs | → | tp | 0.126 * | H1a | Supported | | |
| ccw | → | tp | 0.002 | H1b | Not Supported | H1 | Partially supported |
| co | → | tp | 0.233 ** | H1c | Supported | | |
| cs | → | cp | 0.114 * | H2a | Supported | | |
| ccw | → | cp | 0.066 | H2b | Not Supported | H2 | Partially supported |
| co | → | cp | 0.193 ** | H2c | Supported | | |
| Cs *g | → | tp | −0.118 * | H3a | Supported | | |
| Ccw *g | → | tp | 0.001 | H3b | Not Supported | H3 | Marginally supported |
| Co *g | → | tp | 0.09 | H3c | Not Supported | | |
| Cs *g | → | cp | −0.178 ** | H4a | Supported | | |
| Ccw *g | → | cp | 0.008 | H4b | Not Supported | H4 | Marginally supported |
| Co *g | → | cp | 0.108 | H4c | Not Supported | | |

Note: ccw: commitment to coworkers; co: commitment to organization; cs: commitment to supervisor; tp: task performance; cp: contextual performance; g: generations. * *p* < 0.05, ** *p* < 0.01.

### 3.6. Structural Model

In order to validate the PLS-SEM approach, various quality scores, such as the coefficient of determination ($R^2$) [123], NFI, and SRMR [124], were computed. The $R^2$ values of endogenous constructs are used to evaluate the model fit and indicate how well data points fit a line or curve [123,124]. As suggested by Chin [96], the $R^2$ values are classified as small (0.02 £ $R^2$ < 0.13), medium (0.13 £ $R^2$ < 0.26), or large (0.26 £ $R^2$). The $R^2$ statistic values of the endogenous constructs are used to assess model fit [124,125]. Table 3 shows $R^2$ values as the fit measures of the structural model. According to the outcomes of the main effect model, both task performance ($R^2$ = 0.40) and contextual performance ($R^2$ = 0.28) had small effect sizes. Due to the interaction effect of an innovative climate, $R^2$ for the value of task performance in the final model was 0.10, again reflecting a small effect size, while $R^2$ for the value of contextual performance was 0.13, this time reflecting a medium effect size (see Table 3).

Though the model fit criteria (SRMR, NFI, d_ULS, d_G, and Chi_square) for PLS-SEM apply to the early stages and are often not useful for PLS-SEM, we also report these crucial criteria in this study. The results show that the SRME (standardized root mean square residual) is 0.088 for the main effect model and well above the threshold of 0.080 for the final model (SRMR: 0.097). The NFI (normed fit index) for the main effect model is close to the threshold value of 0.80, while it is far above 0.80 for the final model (NFI: 0.809). Considering the fact that these analyses are difficult to comprehend for the applied subject, the developed model seems to have an overall model fit based on these criteria. Therefore, we can conclude that the structural model developed has quite high predictive power and is satisfactory.

**Table 3.** Structural model.

| Fit Measures | Endogenous Constructs | Main Effect Model | Final Model |
|:---:|:---:|:---:|:---:|
| $R^2$ | tp | 0.072 | 0.076 |
| | cp | 0.101 | 0.131 |
| | SRMR | 0.082 | 0.078 |
| | NFI | 0.773 | 0.809 |

Note: tp: task performance; cp: contextual performance.

## 4. Discussion and Conclusions

Although the effects of organizational commitment on job performance are well studied in the OB literature, generational differences in this relationship in a business environment that has moved to a more remote, online, and technology-based context are still a matter of concern for managers in general, specifically for HRM managers. The consequence of the coexistence of different generations in the workplace is a matter of concern for sustainability in general and the sustainability of employee attitudes and behaviors in particular. In this study, we aimed to define how generational differences between Gen X and Gen Y employees affect the relationship between AC and sustainable job performance since AC can shape and strengthen job performance in the long term. To accomplish this, we developed and found support for a model in which generational differences moderate the relationship between foci of affective commitment (affective commitment to the organization, affective commitment to the supervisor, and affective commitment to coworkers) and job performance (task performance and contextual performance) between Gen X and Gen Y workers in Turkey. Specifically, two of three foci (affective commitment to the organization, affective commitment to the supervisor, and affective commitment to coworkers) were found to be positively related to job performance. Moreover, the relationship between commitment to the supervisor and job performance was weaker for Gen Y than for Gen X. Several meaningful theoretical implications can be derived.

Regarding the implications for theory and practice, the neglect of generational differences in adapting leadership styles to the workforce may risk undermining employee performance, which may lead to a higher turnover rate [126]. For organizations, more attention should be given to generational differences in order to achieve higher commitment and higher employee performance in the long term, that is, in terms of sustainability. From this study, it seems clear that managers should adopt a new leadership style that has greater appeal to younger generations. Gen Yers have difficulty with authority. They are identified as self-confident and even quite selfish. However, they are career- and success-oriented. Without a doubt, these younger employees will not respond well to having a dictating or delegating superior. Instead of traditional leadership styles, today's leaders and supervisors should behave more like coaches who show young employees how to succeed. As Shuang et al. [127] recommend, a more participative, democratic, and empowering style may encourage millennials to perform better, leading to organizational sustainability. By developing and nurturing an empowerment climate, the work motivation and commitment of new generation employees may be enhanced.

Of concern for organizations employing a remote workforce in any fashion has been the challenge faced by employees to develop a sense of organizational membership and commitment and achieve high performance. Social isolation as a result of fully or even partially remote work prevents employees from fully experiencing the informal social situations normally experienced in the workplace, which may lead to a decrease in motivation for Gen X in general and older employees in particular. Gen Yers are the first generation to fully adapt to technology in a digital environment [128]. This generation has a very strong personal connection and harmony with advanced technology, including in the workplace [10]. Gen Xers are different. This large extent of technology and isolation, in addition to less socialization and face-to-face communication, makes them anxious. Many adaptation problems, estrangement, and other negative outcomes seem to be more probable for Gen X and older generations. Therefore, the HRM should place

extra emphasis on these employees. For instance, the HRM may provide remote working orientation programs or online socializing events. On the other hand, the opposite may be true for Gen Y and younger generations. They may even be fond of and motivated by remote working, which provides them more freedom and autonomy in addition to the usage of up-to-date technology. For younger employees, modern technology-based HRM technology can be used to reduce the time, energy, and costs incurred, resulting in fewer complaints and conflicts between employers and employees.

From this study, it seems clear that a mental revolution towards management is also required if organizations intend to utilize the full potential of Gen Yers and even the forthcoming Gen Zers. HRM departments will presumably be mediators between the old generation managers and younger employees. HRM departments may organize training programs for managers to create awareness regarding the different characteristics of new generation employees. HRM departments may also prepare a detailed employee inventory regarding demographic characteristics, including their generational and psychological needs and preferences.

*Limitations and Future Research*

There are several limitations to this empirical research that should be noted for future research. First, the use of cross-sectional data is one of the important limitations of this study. Various questions naturally arise about the causality and purpose of the relationships between variables. Similarly, although the survey method used in this research is seen as a key and growing approach in research on the business environment [129], the method used (only the survey) may not provide objective measurements and results about the flow of information. It is assumed that people who fill out the questionnaires answer with full concentration and 100 percent accuracy. For example, it is unclear how the emotional attachment of different generational cohorts towards the organization, coworkers, and supervisors is developed and reflected in job performance as a dynamic phenomenon in the OB context. Problems associated with cross-sectional data and limited time are among the limitations of the research. In future studies, scientists may consider collecting longitudinal data on the evolution of organizational commitment of different generations.

Second, and most likely the most significant limitation, is that the data are self-reported. Although management and OB researchers are not very fond of using self-reports, they cannot do without them, as the practical usefulness of these measures makes them nearly indispensable in many research contexts [130]. Self-reports may not be as limiting as once they were commonly presumed [131]. The extant literature provides evidence concerning the more accurate estimates of self-reports compared with behavioral measures [132]. The present results provide evidence of relationships among the given variables.

Third, this study followed Lyons and Kuron's [96] generational time-based classification to differentiate between Gen X and Gen Y. However, there is still not a consensus regarding the generational periods in the OB literature. Although many Turkish scholars (e.g., [97,98]) follow the same generational classification since it corresponds to socioeconomic conditions and cultural phenomena of Turkish society—an important coup in 1980 and the introduction of the internet in 2000—a different classification may provide deeper or more appropriate insight into how the organizational commitment of different generations is reflected in job performance. Moreover, future studies may involve baby boomers and Gen Zers for a more comprehensive generational comparison. Furthermore, we did not include generational subgroups, and we did not conduct any generational comparisons based on these subgroups since our dataset did not allow for this; further studies may extend samples to include generational subgroups to provide more in-depth results. Finally, readers should be cautious about generalizing the results since our sample is composed of participants who all have higher education. Independent from the generation, higher education may also be a basis of their system of values, attitudes, expectations, behaviors, and technology-based knowledge and skills. For organizations in general, particularly for those that have adopted more remote working structures, as required during the pandemic,

it will be necessary for future studies to specifically examine a more heterogeneous sample that includes participants with different educational backgrounds to determine if similar results are obtained between generations of workers.

Future researchers may also consider enriching the research model by adding more work-related attitudes to foci of commitment, such as job satisfaction and job involvement, or more behavioral outputs, such as organizational citizenship behavior or counterproductive behaviors.

**Author Contributions:** Conceptualization, A.A.Ç.; data curation, M.K.; formal analysis, E.A.; investigation, A.A.Ç. and M.K.; methodology, A.A.Ç., M.K., V.Ö. and A.G.; supervision, E.A. and V.Ö.; validation, V.Ö.; writing—original draft, M.K. and A.G.; writing—review & editing, A.G. All authors have read and agreed to the published version of the manuscript.

**Funding:** This research received no external funding.

**Institutional Review Board Statement:** Not applicable.

**Informed Consent Statement:** Informed consent was obtained from all subjects involved in the study.

**Data Availability Statement:** Not applicable.

**Conflicts of Interest:** The authors declare no conflict of interest.

## Appendix A. Measures

Standardized loadings are in parentheses.

CR: composite reliability; α: Cronbach alpha, AVE: average variance extracted; $r_{wg}$: inter-rater agreement.

*: denotes a dropped item; either they reduced the AVE to less than 0.50, or they have low loading weights.

**Affective Commitment**

**Commitment to Coworkers** (*Adapted from* Wasti and Can [110])

I do not feel emotionally attached to my coworkers (0.735)

I would be very happy to spend the rest of my professional life working with my current colleagues. (0.834)

I truly feel my coworkers' issues as my own. (0.856)

I do not have a strong sense of belonging to my coworkers. (0.824)

Working with my colleagues has a very special meaning to me. (0.850)

CR = 0.912

α = 0.879

AVE = 0.674

**Commitment to Supervisor** (*Adapted from* Wasti and Can [110])

I do not feel emotionally attached to my supervisor. *

I would be very happy to spend the rest of my professional life working with my current supervisor. (0.906)

I really feel my supervisor's issues as my own. (0.880)

I do not have a strong sense of belonging to my supervisor. (0.675)

Working with my supervisor has a very special meaning to me. (0.654)

CR = 0.864

α = 0.800

AVE = 0.619

**Commitment to Organization** (*Adapted from* Wasti and Can [110])

I would be happy to spend the rest of my professional life in this organization. (0.860)

I truly feel that this organization's issues are like my own. (0.884)

I do not have a strong sense of belonging to my organization. *

I do not feel emotionally attached to this organization. (0.713)

This organization has a very personal (special) meaning to me. (0.846)

CR = 0.897

α = 0.857

AVE = 0.686
**Task Performance** (*Adapted from* Goodman and Syvantek [112])
Achieves the objectives of the job. (0.806)
Meets criteria for performance. (0.782)
Demonstrates expertise in all job-related tasks. (0.758)
Fulfills all the requirements of the job. (0.824)
Could manage more responsibility than typically assigned. (0.753)
Appears suitable for a higher-level role. *
Is competent in all areas of the job, handles tasks with proficiency. (0.739)
Performs well in the overall job by carrying out tasks as expected. (0.882)
Plans and organizes to achieve objectives of the job and meet deadlines. (0.767)
CR = 0.930
α = 0.915
AVE = 0.624
**Contextual Performance** (*Adapted from* Jawahar and Carr [113])
Helps other employees with their work when they have been absent. (0.711)
Volunteers to do things not formally required by the job. (0.679)
Takes initiative to orient new employees to the department even though not part of his/her job description. (0.800)
Helps others when their work load increases (assists others until they get over the hurdles). (0.814)
Assists me with my duties. (0.841)
Makes innovative suggestions to improve the overall quality of the department. (0.720)
Willingly attends functions not required by the organization, but helps in its overall image. (0.706)
CR = 0.902
α = 0.873
AVE = 0.570

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
