# Peer review of "Does the Reflection of Foci of Commitment in Job Performance Weaken as Generations Get Younger? A Comparison between Gen X and Gen Y Employees"

_sustainability, doi:10.3390/su13169271_

Round 1
Reviewer 1 Report
Relevant and interesting but over-discussed topic, the study bases on a strong and appropriate econometrical and statistical methodology but suffers from the lack of fresh and new literature, mainly in these extraordinary times (please let me remark here, that the authors might refer to that how the COVID-19 has affected these generations' role in the OB, e.g. foci of commitment via online).
First of all, please provide a correct definition and age range of who Turkish Millennials and GenX are. The generations are age groups born in the same era and thus socialized in a similar environment BUT cultural differences must be taken into account because some chronological schemes and impacts reach the cultures and regions (countries) differently at different times. Of course, as the authors also underlined, it does not mean that all members of the same age group are identical. Moreover, as Kolnhofer et al. (2018)* proved, there are many characteristics that are age and life period-specific (see development psychology) and cannot be only reasoned by the generation differences. That is why Generation X is (incorrectly) described as family-oriented. Time is flying and there will be a life period in the Millennials' weekdays when they are having and raising babies which requires more "family-oriented" specifics. Here, if the dataset allows, it might worth comparing answers regarding other demographical status. Although, researchers are working with clear age ranges (in order to make comparison easier), the so-called between generations subgroup should be also taken into account.
*Kolnhofer Derecskei, A. - Reicher, R. Zs. - Szeghegyi, Á. (2018) The X and Y Generations' Characteristics Comparison. Acta Polytechnica Hungarica 14(8):2017-107 DOI: 10.12700/APH.14.8.2017.8.6
Reviewer 2 Report
The question for authors: how do you describe the sustainable job performance and what difference is between job performance and sustainable job performance? It is understandable that the authors want to fit to the scope of the journal, but may be you can find another means for that.
I am not sure if the word foci is adequate in the context in which it is used.
Methods: you measured perceived or subjective job performance and you should very clearly inform about that in your paper. The instrument which you have used did not let to evaluate objective job performance. You can find a lot of scientific literature about these two different methods of measurement of job performance (the results of the research can be different regarding the method which you use for evaluating job performance) and you need to talk about this specific peculiarity of the measurement not only in methodological part of manuscript but also in the theoretical backround. It is not enough to write in Limitations that data was self-reported because everything talking about such behaviour like job performance is much more complicated.
line 330: effective form - what does it mean?
Table1: I have doubt if there are not mistakes in the table, for example, what does it mean ccw correlation with ccw 0.821 or cp correlation with cp?
The lack of information is to understand the results of hypothesis. More detailed information on the calculations performed should be provided.
Discussion should be strengthen with deeper analysis of the results.
Limitations: the sample is very specific (all participants are with higher education) and you need to discuss about that and how this fact could influence the results.
Please, be more careful regarding text and references. For example, you are writing Wasti and Can in one sentence and Waste and Can in the next one. There are more editorial type mistakes in the manuscript which should be corrected.
All abbreviations in the text should be explained, e.g., what OC; OB means.
Reviewer 3 Report
I believe that the paper examines an interesting relationship between commitment and performance, which is extensively analyses in literature, but needs more evidence regarding the interaction of variables (SEM methods). In the abstract and the introduction I would like to see the value of examining the differences among generations X and Y, why is important and what are the implications of this result in the moment of raising this research question?
Literature review and hypotheses development are were built, but the use of alphabetical list a), b), c) cannot convince me, I would prefer the hypothesis only with text. Example:
Relationships between affective commitment a) affective commitment to supervisor, b) affective commitment to co-workers, and c) affective commitment to organization and task performance are moderated by generational differences in a way that commitment-performance relationships are weaker for GenYs than GenXs.
Relationships between affective commitment towards supervisor, co-workers, organization and task performance are moderated by generational differences in a way that commitment-performance relationships are weaker for GenYs than GenXs.
Methods and analysis are well developed. The sections of conclusions and limitations also are well elaborated. I will miss from my point of view a section of recommendations (practitioners), it will interesting to know how to provide motivation for workers of different generations.
It's a good and well structured paper.
Round 2
Reviewer 2 Report
Thank you for corrections of the manuscript.